

# Synoptic-scale conditions and convection-permitting hindcast experiments of a cold-season derecho on 3 January 2014 in Western Europe

Luca Mathias[1], Patrick Ludwig[2], Joaquim G. Pinto[2]

[1]Institute for Geophysics and Meteorology, University of Cologne, Cologne, Germany
[2]Institute of Meteorology and Climate Research, Karlsruhe Institute of Technology, Karlsruhe, Germany

*Correspondence to*: Luca Mathias (luca.mathias@outlook.com)

**Abstract.** A major linear mesoscale convective system caused severe weather over northern France, Belgium, the Netherlands and northwestern Germany on 3 January 2014. The storm was classified as a cold-season derecho with widespread wind gusts exceeding 25 m s$^{-1}$. While such derechos occasionally develop along cold fronts of extra-tropical cyclones, this system formed in a postfrontal air mass along a baroclinic surface pressure trough favoured by strong large-scale air ascent induced by an intense mid-level jet. The lower-tropospheric conditions were characterized by weak latent instability and strong vertical wind shear. Given the poor operational forecast of the storm, we analyse the role of initial and lateral boundary conditions to the storm's development by performing convection-permitting simulations with different datasets. The storm is best represented in simulations with high temporally and spatially resolved ERA5 initial and lateral boundary conditions, which provide the most realistic development of the essential surface pressure trough. Moreover, simulations at convection-resolving scale enable a better representation of the observed derecho intensity. This case study indicates that high resolution ensemble simulations might be important to overcome the current shortcomings of forecasting cold-season convective storms, particularly for cases not associated with a cold front.

## 1 Introduction

Mesoscale convective systems (MCSs) often affect Central Europe, particularly in late spring and summer. In some cases, MCSs can exhibit a linear structure, endure for several hours and lead to both intense wind gusts and precipitation over large areas, and are sometimes classified as derechos (Johns and Hirt, 1987). While such events primarily occur over Western Europe during the summer half year (Gatzen, 2004), they might also occur during wintertime (Gatzen et al., 2011). The majority of such cold-season derechos occur in association with the passage of a cold front from an extra-tropical cyclone, which is embedded in a northwesterly flow (Ludwig et al., 2015; Gatzen, 2018). However, on 3 January 2014, a linearly organised convective system formed not along a cold front, but in a postfrontal air mass within a southwesterly flow and crossed over the northern tip of France, the Benelux region and the northwestern part of Germany, causing severe straight-line wind damage along an approximately 650-km-long path (Fig. 1). The magnitude of the convective gusts ranged mostly



between 20 to 30 m s$^{-1}$, but hurricane-force wind gusts (> 32.7 m s$^{-1}$) were measured locally between 1300 and 2200 UTC (Fig. 1). Additionally, F1-rated wind damage was reported in western Belgium and northwestern Germany (Fig. 1). According to these observations, this convective event can be classified as a cold-season derecho following the definition of Johns and Hirt (1987). Moreover, three tornadoes have been confirmed and several damage locations were under suspicion of being produced by further tornadoes according to the European Severe Weather Database (ESWD; Dotzek et al., 2009). In

addition to the nontornadic and tornadic wind damage, local reports of thick layers of small hail or graupel are archived in the ESWD. Furthermore, the derecho-producing mesoscale convective system (DMCS) was not well anticipated by the national weather services. The short-term synoptic reports by the German Weather Service [Deutscher Wetterdienst (DWD)] and the Royal Netherlands Meteorological Institute [Koninklijk Nederlands Meteorologisch Instituut (KNMI)], issued in the morning of 3 January 2014, mentioned the probability of isolated strong thundery showers with the risk of storm-force wind

gusts in the afternoon and evening. The online report[1] by the European Storm Forecast Experiment (ESTOFEX) pointed out the potential for the development of a convective line that could cause severe winds and isolated tornadoes in the Netherlands. However, the forecast level 1 threat area issued by ESTOFEX did not cover the main region that was affected by the long-lived convective system. These specific characteristics motivate a detailed review of this event.

Most of the studies dealing with the environmental conditions, climatology and modelling of DMCSs originate from the

United States. It is pointed out that the large-scale conditions associated with derecho events are highly variable (e.g., Evans and Doswell, 2001; Coniglio et al., 2004; Cohen et al., 2007). DMCSs developing in strongly forced synoptic regimes are associated with weak latent instability [i.e., low values of convective available potential energy (CAPE)] and high shear values, which is mostly the case during the cold season (e.g., Bentley and Mote, 2000; Evans and Doswell, 2001; Gatzen et al., 2011). In addition, cold-season derechos sometimes occur in environments of very limited low-level moisture (i.e., 2 m

AGL dew points below 10°C), which are then referred to as low-dew point derechos (Corfidi et al., 2006). The high-shear, low-CAPE (HSLC) environments are very challenging with regard to the operational forecast of severe convection (Sherburn and Parker, 2014a,b).

Nevertheless, efforts have been made since the mid 2000's towards a better understanding of European derechos (e.g., Gatzen, 2004; Punkka et al., 2006; Lòpez, 2007; Gatzen et al., 2011; Hamid, 2012; Celiński-Myslaw and Matuszko, 2014;

Toll et al., 2015). Gatzen (2018) identified and classified 40 derechos that affected Germany during the 18-year period 1997-2014, including 12 winter cases. However, modelling studies about European derechos are rarely found in the literature. For instance, Toll et al. (2015) performed hindcast experiments of a warm-season derecho in Northeastern Europe and Ludwig et al. (2015) were able to successfully reproduce the derecho intensity of deep convection associated with the cold front of winter storm Kyrill in 2007 (Fink et al., 2009). Hence, more observational and numerical studies about well-organised

DMCSs developing in cold season situations are needed, for instance to better understand the processes and potentially enhance the predictability of these uncommon events.

---

1    http://www.estofex.org/cgi-bin/polygon/showforecast.cgi?text=yes&fcstfile=2014010406_201401030002_1_stormforecast.xml



The purpose of this study is to analyse the synoptic characteristics and the predictability of this derecho event. With this aim, we examine the presence of the ingredients necessary for the development of the severe cold-season DMCS. In situ observations and numerical weather prediction (NWP) model data enable a detailed examination. Given the poor
performance of the operational forecasts, high-resolution hindcast experiments are performed to investigate the reasons for this shortcoming.

This article is structured as follows. Section 2 describes the data and methods. The synoptic-scale situation and the environmental conditions associated with the convective windstorm are highlighted in section 3. Section 4 discusses the predictability issues and analyses the model experiments. The last section includes a short summary and our conclusions.

**2 Data and numerical model**

The in situ wind measurements used in this study include data from the synoptic weather station networks operated by numerous national weather services [Météo-France, Royal Meteorological Institute of Belgium (RMIB), United Kingdom's Meteorological Office (UK Met Office), KNMI, DWD] and by the private weather service MeteoGroup. The 1200 UTC upper-air sounding from Larkhill (WMO 03743) is considered as representative for the environmental conditions in which
the DMCS developed. The processing and visualization of the upper-air data was done with the Rawinsonde Observation Program (RAOB) for Windows. Additionally, radar composites supplied by the RMIB and KNMI are processed with MATLAB version R2016a. The RMIB composite image is produced on a 500-m grid by combining pseudo Constant Altitude Plan Position Indicators (CAPPI) at 1.5 km altitude of four operative C-band radars located in Belgium and France. The KNMI composite image consists of pseudo CAPPI at a height of 1.5 km on a 1-km grid, which are based on the
measurements of two Dutch C-band Doppler radars. In addition to the in situ and radar data, the recently released ERA5 data from the European Centre for Medium-Range Weather Forecasts (ECMWF) are used to examine the synoptic-scale conditions. ERA5 was produced using 4DVar data assimilation with the model cycle Cy41r2 of ECMWF's Integrated Forecast System (IFS). The hourly reanalysis data output has a grid interval of approximately 31 km (Hersbach and Dee, 2016). Furthermore, the predictability issue will be briefly described using the operational ECMWF's Ensemble Prediction
System (ECMWF-EPS) and the Consortium for Small-scale Modelling Limited-area Ensemble Prediction System (COSMO-LEPS). ECMWF-EPS consists of 50 perturbed members and one control run with a grid spacing of about 32 km (IFS release Cy40r1). COSMO-LEPS includes 16 ensemble members with a grid interval of approximately 7 km. The initial and boundary conditions for each of these 16 members are selected based on a cluster analysis from two consecutive ECMWF-EPS runs (Montani et al., 2011).

The COSMO model (version 5.0, subversion 9) is used in its climate version (CLM), henceforth termed CCLM (Rockel et al., 2008), to perform high-resolution hindcast simulations of the event. The CCLM is synchronized regularly with the NWP version of the COSMO model operationally used at the DWD, but excluding data assimilation or latent heat nudging. The CCLM has shown its capabilities in several convection-permitting modelling studies in the recent past (e.g., Fosser et al., 2015; Ludwig et al., 2015; Leutwyler et al., 2016; Mathias et al., 2017). For this study, a total of three simulations (each





including several nesting steps) have been realised to analyse the DMCS in more detail. A reference simulation is driven by

initial and boundary conditions taken from the ERA5 dataset. Additional hindcast experiments have been conducted using

ERA-Interim reanalysis (ERAI, IFS release Cy31r2; Dee et al., 2011) and ECMWF operative analysis data (ECAN, IFS

release Cy40r1) to investigate the sensitivity of different initial and boundary conditions on the DMCS development. Besides

the different data assimilation cycles, both datasets differ in their grid spacing (ERAI: T255, $\Delta x \approx 80$ km; ECAN: T1279, $\Delta x$

$\approx 16$ km) and their temporal resolution (hourly data for ERA5, 6-hourly data for ERAI and ECAN) which in the end leads to

two additional distinct input datasets.

A three-step nesting approach is necessary to obtain a very fine grid spacing ($\Delta x \approx 1.1$ km) in the ERA5-driven reference

simulation. The ERA5 and ECAN data are first downscaled over domain 1 (D1) with a horizontal grid interval of 7 km,

followed by domain 2 (D2, $\Delta x \approx 2.8$ km) and finally domain 3 (D3, only for ERA5, $\Delta x \approx 1.1$ km; see Fig. 2 for domain

configuration). For ERAI initial and boundary conditions, an additional preceding nesting step (D0, grid spacing of 25 km) is

necessary to avoid large resolution jumps (Matte et al., 2017). The ERAI and ECAN simulations are both downscaled to a

final grid interval of 2.8 km (D2) in order to analyse the differences in the atmospheric conditions during the development of

the DMCS in comparison to the ERA5 reference simulation. The 1.1-km simulation forced with ERA5 data is used for a

detailed comparison with radar and wind gust observations.

The CCLM is able to resolve deep moist convection (convection-permitting model; Prein et al., 2015) at grid intervals

smaller than 4 km, while shallow convection is still parameterised. Thus, for the first nesting steps (D0, D1) the convective

mass flux is parameterised after Tiedtke (1989), while for the higher resolution runs (D2, D3) this scheme is only applied to

shallow convection (see Table 1). The wind gusts are estimated based on a diagnostic parameterisation depending on the

wind speed at 10 m AGL and the friction velocity (Schulz, 2008):

115                                      $$v_g = v_{10\,m} + 3.0 \cdot 2.4 \cdot u^* ,        \qquad (1)$$

with the empirical factors 3.0 and 2.4 motivated by the Prandtl layer theory (Panofsky and Dutton, 1984). The friction

velocity is computed using the drag coefficient for momentum $C_D$ and the wind speed at 10 m AGL:

$$u^* = (C_D)0.5 \cdot v_{10\,m}        \qquad (2)$$

An overview of the physical parameterisations that are used for all domains is given in Table 1 and a more detailed

description can be found in Doms et al. (2011). To overcome unbalanced information for the mass and wind field in the

initialization process and to accelerate the spin-up process, a time filtering approach after Lynch (1997) is applied in CCLM.

The ERA5-driven reference simulation of the event over D1 is initialised at 0000 UTC on 3 January 2014, approximately 12

hours before the DMCS developed. The higher resolution simulations start at 1200 UTC (D2) and 1300 UTC (D3),

respectively. The hindcast experiment with ERAI is initialised at 0600 UTC (D0) on 3 January 2014 with subsequent nesting

steps at 0900 UTC (D1) and 1200 UTC (D2). For the ECAN data, the hindcast experiment has been initialised at 0600 UTC

(D1) and subsequently at 1200 UTC (D2). The aforementioned temporal setup is used to permit a most realistic simulation of

the derecho and allow a direct comparison of the simulated data with the observational data. ECAN- and ERAI-driven

simulations with an identical starting time on D1 (as for ERA5, 0000 UTC) have also been computed, but they will not be





further discussed here due to poorer performance. Furthermore, additional simulations have been conducted to analyse the
sensitivity of initial and lateral boundary conditions (ILBCs) on the resulting derecho. Regarding the initial conditions,
ECAN-driven simulations initialised at 0000 UTC were performed with the initial ERA5 wind and moisture fields, while the
ECAN boundary conditions remained unchanged. Regarding the lateral boundary conditions (LBCs), an additional ERA5-
driven simulation was performed in which LBCs are updated every 6 hours (as opposed to hourly updates in the reference
simulation). For all experiments, the model output is stored on hourly basis for the 7-km simulations and with a 15-minute
interval for the 2.8-km and 1.1-km simulations.

## 3 Synoptic-scale overview and storm environment

The large-scale environmental conditions associated with the derecho are examined based on ERA5 reanalysis data and
upper-air soundings. At 1200 UTC on 3 January 2014, an intensive low pressure system (core pressure of 949 hPa) named
"Anne" was situated over the Northern Atlantic close to Scotland (Fig. 3) and high pressure (1022 hPa) was analysed north
of the Alps. Consequently, a strong horizontal pressure gradient existed over the British Isles and over parts of France,
Belgium and the Netherlands. The frontal system of the surface low extended from the Norwegian Sea over Denmark and
Germany all the way south to the Iberian Peninsula (Fig. 3). The occluded front had a warm character, meaning that the near-
surface air directly behind the front was slightly warmer and moister than the prefrontal air (not shown). Moreover, a surface
trough or discontinuity line was analysed by the UK Met Office over the English Channel (Fig. 3), which was related to the
development of the DMCS. At 1500 UTC, the surface trough reached western Belgium and corresponded to a weak
isallobaric gradient (Fig. 4a). This trough was associated with large-scale upward motion located at the cyclonic exit of a
mid-level jet (Figs. 4c,e), with a convergence maximum at about 900 hPa and a divergence maximum at roughly 475 hPa
(Figs. 5a). In addition, the pressure trough was associated with weak baroclinity, because the lower-tropospheric temperature
dropped by a few Kelvin after the passage of the trough (Fig. 5a). Three hours later at 1800 UTC, the surface trough was
located over northwestern Germany and the isallobaric gradients strengthened (Fig. 4b). The pressure decrease ahead of the
trough was however much smaller than the following pressure rise (-0.4 hPa h$^{-1}$ vs. +1.2 hPa h$^{-1}$). The trough also remained
in phase with the large-scale forcing for ascent, as it was vertically aligned with strong divergence at the exit of the mid-level
jet and ahead of a negatively tilted upper-level trough situated over Belgium (Figs. 4d,f and 5b).

The ingredients-based method by Johns and Doswell (1992) prescribes three necessary elements for the occurrence of deep
moist convection. First, a sufficient amount of moisture in the boundary layer is required. For the analysed event, a tongue of
enhanced low-level moisture existed between the occluded front and the postfrontal surface trough at 1200 UTC (cf. Fig. 3a
and Fig. 6a). Near-surface dew points of 7 to 9 °C (not shown) and 950 hPa specific humidity values above 5 g kg$^{-1}$ were
observed over France, western Germany and the Benelux region (Fig. 6a). Backward trajectories suggested that the unusual
moist air mass (for this season) was advected from the Northeastern Atlantic over the Bay of Biscay towards Western
Europe (not shown). At 1800 UTC, the moisture tongue covered eastern France and large parts of Germany with slightly
lower values of specific humidity (Fig. 6b).

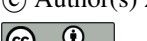



The second necessary ingredient is a sufficiently steep lapse rate in the lower to middle troposphere above the moist layer. At 1200 UTC, lapse rates of 6.5 to 7 K km$^{-1}$ between 900 and 650 hPa covered the British Isles, the English Channel and northwestern France (Fig. 6c). Upper-air observations revealed a conditionally unstable air mass confined to the layer below

650 hPa with a striking capping inversion between 650 and 600 hPa (e.g., at Larkhill; see Fig. 7), which was induced by the subsiding air from a potential vorticity intrusion (Gatzen, 2018). The combination of steep lapse rates and low-to-moderate boundary layer moisture resulted in low CAPE values of 150 to 200 J kg$^{-1}$, as indicated by the 1200 UTC sounding from Larkhill (Fig. 7). At 1800 UTC, this area of weak latent instability reached northwestern Germany (Figs. 6b,d).

Finally, the vertical wind shear is a crucial ingredient for linearly organised MCSs, as shown in various numerical and

observational studies (e.g., Weisman and Klemp, 1982; Rasmussen and Blanchard, 1998). In this case, the DMCS formed in an environment with 0-6 km bulk shear values well above 25 m s$^{-1}$ (Fig. 6a). The 1200 UTC sounding from Larkhill also revealed almost unidirectional 0-6 km bulk shear and mean wind speed values of about 30 m s$^{-1}$ (Fig. 7). Thus, the deep layer shear and mean wind vector were nearly parallel, which favoured the development of a fast downwind-propagating and severe MCS (Corfidi, 2003; Cohen et al., 2007). The lower-tropospheric shear was also very strong with 0-3 km bulk shear

values larger than 15 m s$^{-1}$ (Fig. 6e). According to ERA5 reanalysis data, these shear magnitudes remained more or less constant at 1500 UTC over Belgium and at 1800 UTC over northwestern Germany (Fig. 6f). The lifting mechanism, as the last indispensable ingredient, was provided by the surface pressure trough and the associated low-level convergence.

In brief, the derecho on 3 January 2014 developed in a strongly forced synoptic regime, which was associated with a baroclinic surface trough (Sanders, 1999; Sanders, 2005). The DMCS evolved within an area characterized by a) a sufficient

amount of lower-tropospheric moisture, b) steep lower-tropospheric lapse rates of 6.5 to 7 K km$^{-1}$, c) weak latent instability (CAPE < 200 J kg$^{-1}$) and d) strong vertical wind shear, with the majority of the shear and latent instability located in the lowest 3 km of the troposphere. This HSLC environment generally allows the formation of cold-season DMCSs producing severe winds, especially in presence of strong large-scale forcing for ascent (e.g., Bentley and Mote, 2000; Evans and Doswell, 2001). In comparison with two other European cold-season derechos studied by Gatzen et al. (2011), this event was

characterized by much weaker vertical wind shear. For example, the Kyrill derecho formed in an environment with 0-6 km bulk shear values of up to 65 m s$^{-1}$ (vs. 30 m s$^{-1}$ for this case). Similarities were found among the magnitude of low-level specific humidity and lower-tropospheric lapse rates (Gatzen, 2018).

**4 Predictability and high resolution modelling**

Model hindcast experiments are used to complement the description of this extreme cold-season convective event. The

following subsections include a short analysis of the operational ensemble forecasts and a detailed examination of the differences between the ERA5-, ERAI- and ECAN-driven CCLM simulations. Furthermore, the benefit of our highest-resolution simulation will be highlighted in the last subsection.

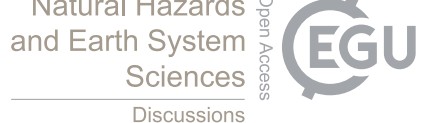

### 4.1. Ensemble forecasts

As already mentioned in the introduction, the DMCS on 3 January 2014 was not well forecasted. The probabilistic forecast
issued from the ECMWF-EPS 0000 UTC run revealed a probability of 40 to 60 % for the occurrence of maximum surface
wind gusts exceeding 20 m s$^{-1}$ over Belgium on 3 January 2014 and a much lower probability for western Germany (Fig. 8a).
The predicted likelihood for gusts reaching wind speeds larger than 25 m s$^{-1}$ was zero for the whole investigation area,
except for the marine areas of the English Channel and the North Sea (Fig. 8c). COSMO-LEPS provided similar
probabilistic forecasts (Figs. 8b,d) and showed even a lower probability for wind gusts exceeding 20 m s$^{-1}$ over northwestern
France, Belgium and western Germany than ECMWF-EPS (cf. Figs. 8a and 8b).

### 4.2. Dependence on initial and lateral boundary conditions

To investigate the potential predictability of the derecho event, CCLM hindcasts were performed using the ERA5, ERAI and
ECAN data as ILBCs. The ERA5-driven CCLM simulation (CCLM-ERA5) revealed a linearly organised convective system
over parts of northern France, Belgium and the North Sea at 1600 UTC, which was associated with a convergence zone
along a surface pressure trough (Figs. 9a,b). In the ERAI-driven CCLM simulation (CCLM-ERAI), deep moist convection
formed in a similar way, but the surface trough was slightly displaced to the north and the convective cells remained initially
mostly discrete (Figs.9c,d). At a later time step in this simulation (2000 UTC), a linearly organised MCS became apparent
(not shown). The ECMWF operative analysis driven simulation (CCLM-ECAN) developed discrete and non-severe
convective cells over the investigation area along unorganised near-surface convergence zones, as no well-defined surface
pressure trough was evident in this simulation (Figs. 9e,f). In general, the CCLM-ECAN simulation is clearly distinct from
the results with ERA5 and ERAI reanalysis boundary conditions, despite that no major differences in the simulation of
CAPE could be identified (cf. Fig. 9f with Figs. 9b and 9d). All three simulations feature maximum CAPE values of 200 to
250 J kg$^{-1}$ over the Netherlands (not shown). Apparently, the differently simulated structure of the convection-initiating
boundary had a major impact on the subsequent upscale growth of the convection.  In general, both CCLM-ERA5 and
CCLM-ERAI simulated a nearly closed convergence band in contrast to CCLM-ECAN (cf. Figs. 9a and 9c with Fig. 9e).
Even exchanging the initial specific humidity and wind fields in CCLM-ECAN with ERA5 values did not result in
significant improvements. However, a considerable sensitivity was found when modifying the update frequency of the LBCs
in CCLM-ERA5: The ERA5-driven CCLM simulation with 6-hourly LBCs did not simulate the surface pressure trough
associated with the development of the DMCS, which extends from southeastern England to northern France at 1400 UTC in
the reference simulation with hourly LBCs (Figs. 10a,b). The absence of this trough resulted in a weaker and less organised
convective system (not shown). To determine the cause for the missing trough, we investigated the synoptic-scale
differences between both CCLM-ERA5 simulations at the western boundary of the model domain D1. A striking pressure
anomaly entered D1 from the west between 0700 and 0900 UTC, which had its origin in an additional surface pressure
trough located west of Ireland in CCLM-ERA5 with hourly LBCs (Figs. 10c,d). We hypothesize that this trough affected the





pressure field downstream over the English Channel, leading to the formation of the pressure trough associated with the derecho between 1200 and 1500 UTC in the ERA5-driven reference simulation. We thus propose that the realistic representation of the convection-initiating boundary and the associated low-level forcing for ascent, which was achieved with initial ERA5 data and hourly LBCs, would have been the key factors to successfully forecast this cold-season storm.

**4.3. CCLM-ERA5 1.1-km simulation and comparison with the observed event**

As the CCLM-ERA5 simulations revealed a good representation of the DMCS in terms of its spatiotemporal evolution, this subsection will include a detailed analysis of the system using the highest-resolution run.

Between 1300 and 1400 UTC, several convective cells initiated over northern France and the English Channel along two distinct low-level convergence zones (cf. Figs. 11a and 11b). Both convergence zones were associated with isallobaric gradients (see yellow dashed lines in Fig. 11a) and a weak gradient of equivalent potential temperature in 850 hPa (not

shown). Since the 0-6 km mean wind vector had a large component perpendicular to the convection-initiating boundaries (not shown), the convective cells over northern France remained mostly discrete and their upscale growth was initially limited. While the convective cells moved towards the northeast, they were subjected to weak latent instability (CAPE < 300 J kg$^{-1}$; see Fig. 11b). At 1600 UTC, the convective cells organised and merged to a linearly organised MCS extending from the North Sea over the Benelux region to northern France (Fig. 11c), as both convergence zones phase locked along the

surface pressure trough (Fig. 9a). Still, the MCS benefits from low-end CAPE (< 150 J kg$^{-1}$) downstream of the system (Fig. 11c). At 1900 UTC, the simulated DMCS reached western Germany exhibiting its peak organisation (Fig. 11d). As the linear storm system moved farther east into an environment with a drier and colder boundary layer, it began to weaken (decreasing reflectivity) and gradually lost its organisation after 2030 UTC due to the lack of latent instability (not shown). Compared to the evolution of the observed DMCS, the CCLM-ERA5 run featured a broken-line mode of the DMCS, especially during the

early stage of the system's life cycle (cf. Figs. 12a and 12b). However, the bowed or hooked segments observed in the real case (see Figs. 12a and 12c) were also present in the highest-resolution simulation, for example at 1600 UTC over central Belgium (50.25°N 4.5°E, Fig. 10c) or at 1900 UTC over northwestern Germany (52°N 8.5°E, Fig. 11d). The dissipation phase of the DMCS was also realistically simulated by CCLM.

The maximum wind gust pattern obtained from CCLM-ERA5 shows some striking differences among the 2.8-km and 1.1-

km simulations. The former shows multiple stripes of gusts ranging between 20 and 30 m s$^{-1}$ over the onshore areas, with a single local wind maximum of about 35 m s$^{-1}$ over northeastern Netherlands (Fig. 13a). By contrast, the highest-resolution simulation covers a larger area with convective gusts exceeding 20 m s$^{-1}$, which matches well with the observations (cf. Figs. 1 and 13b). In addition, the 1.1-km simulation highlights the potential for hurricane-force gusts much better, exhibiting local maxima of up to 45 m s$^{-1}$ over the mountainous regions of eastern Belgium, but also over the lowlands of northern Germany

(Fig. 13b). This shortcoming of the 2.8-km simulation is probably linked to a less accurate representation of the convective-scale processes, since it has a significant lower vertical resolution than the 1.1-km simulation (see Table 1). More precisely, the downdrafts of the individual convective cells are slightly stronger in the 1.1-km simulation, leading to stronger pressure




gradients along their gust fronts compared to the 2.8-km simulation (cf. Figs. 13c and 13d). As the computation of the horizontal wind is affected by the pressure gradient force, the friction velocity will increase due to higher horizontal wind
speeds, which will result in stronger gusts following Eqs. (1) and (2).

Overall, we demonstrated that simulations with a grid spacing of about 1 km are necessary to realistically approach the severity of deep moist convection within the HSLC environment on 3 January 2014. However, Ludwig et al. (2015) were able to viably reproduce the observed gust intensity of the European derecho on 18 January 2007 using a coarser grid interval of 2.8 km (see Figs. 8d-f and 12 in Ludwig et al., 2015). The main difference between both simulations is the linear
upscale growth of the simulated convection. The DMCS modelled by Ludwig et al. (2015) featured a nearly closed narrow convective line along Kyrill's cold front, which is in contrast to the less organised DMCS of the CCLM-ERA5 simulation in the present study. This disparity is most likely attributable to the nature of the convection-initiating boundary (cold front vs. baroclinic trough). Furthermore, the synoptic background flow was stronger during the Kyrill derecho. Thus, we speculate that the magnitude of the simulated wind gusts might be sensitive to the convective upscale growth along the convection-
initiating boundary when using convection-permitting CCLM configurations with coarser grid spacing.

## 5 Summary and conclusions

In this study we have analysed the synoptic characteristics and the predictability of a major linear mesoscale convective system which developed in a postfrontal air mass and caused severe weather in northern France, Belgium, the Netherlands and northwestern Germany on 3 January 2014. The system produced hurricane-force winds and was classified as a moderate
low-dew point derecho as it satisfies the criteria of Johns and Hirt (1987), Coniglio and Stensrud (2004) and Corfidi et al. (2006). Cold-season derechos that are not associated with a cold front are uncommon in Germany (Gatzen, 2018).

First, we have investigated the environmental conditions in which this DMCS developed, revealing that the system formed in a strongly forced synoptic regime marked by a strong southwesterly upper-level flow. In particular, the DMCS benefited from large-scale forcing for ascent since it was positioned at the left exit of a strong mid-level jet, which is typical for
European cold-season derechos (Gatzen, 2018). The formation of the DMCS was also associated with a baroclinic surface pressure trough in the postfrontal air mass. Moreover, the DMCS evolved in an environment that featured the three necessary ingredients for the occurrence of deep moist convection (Johns and Doswell, 1992). Steep lower- to mid-tropospheric lapse rates and enhanced amounts of boundary layer moisture could be identified. The resulting weak latent instability was mostly concentrated within the lowest 3 km of the troposphere, in which the strongest vertical wind shear was also present.
However, the tropospheric speed shear was much weaker in contrast to cold-season derechos developing along a cold front (Gatzen et al., 2011). This lower-tropospheric HSLC regime, in combination with low-level convergence along the surface trough, may have been crucial for the linear organisation of the DMCS and for the development of bowing line segments, which were observed in radar imagery (Figs. 9a,c).

The analysis of NWP model data revealed the poor performance of the operational forecasts. Thus, high-resolution
numerical experiments (with up to 1.1-km grid spacing) were performed to investigate the reasons for this shortcoming. Our





results provide evidence that the derecho event on 3 January 2014 was predictable given the correct initial and boundary conditions. The ERA5-driven CCLM simulation with hourly updated LBCs produced a linearly organised MCS, whose timing, track and intensity coincided well with the development of the observed DMCS. However, our additional simulations with ERAI and ECAN data as initial and boundary conditions revealed that the development of the storm was sensitive to the

structure of the convection-initiating boundary, which depended on the simulated pressure field. In particular, the simulation with ECAN ILBCs failed to reproduce an organised convective system over the affected region, pointing to a possible shortcoming of the observational analysis in such strongly convective situations (cf. also Mathias et al., 2017). Additional sensitivity experiments revealed the importance of temporal high-resolution LBCs on the development of the DMCS. An ERA5-driven simulation with 6-hourly LBCs performed worse with regard to the intensity and the degree of organisation of

the convection. The reason for this was most likely the absence of the key precursor, a surface pressure trough which entered the model domain between 0700 and 0900 UTC when considering hourly LBCs.

Moreover, we showed that very high horizontal and vertical resolutions were necessary to reproduce the derecho intensity of the simulated convection. This is partially in contrast to the case modelled by Ludwig et al. (2015), which could represent the strong convection embedded in the cold front from storm Kyrill with a coarser grid interval of 2.8 km. However, a higher

model resolution might not always be necessary for a good representation of DMCSs due to the strong case to case variability (Gatzen, 2018), but it might be needed for systems in some cases. Overall, the 3 January 2014 derecho event revealed the difficulty to forecast cold-season convective windstorms when they are not associated with a well-defined synoptic-scale cold front, where upward motion is generally given per se. Therefore, convection-permitting ensemble prediction systems might be considered to improve the predictability of such low probability, high impact events in the

future. Future work will focus on a detailed analysis and high-resolution modelling of other DMCSs affecting Western Europe based on the database established by Gatzen (2018), and test the sensitivity to the ingredients, particularly in terms of the physical mechanisms leading to the large-scale ascent needed to initiate the event.

**Acknowledgments**

We thank the ECMWF for the provision of ERA5, ERA-Interim and ECMWF analysis data. We thank the RMIB and KNMI

for providing radar data. We thank the German Climate Computer Center (DKRZ, Hamburg) for computing and storage resources within the context of DKRZ project ANDIVA (No. 105). We thank Christoph Gatzen for the useful and extensive discussions. We are grateful to the European Severe Storm Laboratory (ESSL) for the reports taken from the European Severe Weather Database (ESWD; www.eswd.eu) shown in Fig. 1. JGP was partially funded by the AXA Research Fund and PL was partially funded by REKLIM.



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





**Figure captions**

**Figure 1.** The radar-observed position of the leading edge of the derecho-producing mesoscale convective system at hourly intervals between 1300 and 2200 UTC on 3 January 2014 is shown by the dashed black lines. The observed maximum wind
gusts (m s$^{-1}$) are denoted by the small colored squares (see legend). The white edged squares indicate gusts stronger than 25.7 m s$^{-1}$. Tornadic and nontornadic wind damage locations are marked by the small blue and magenta triangles, respectively (see legend). The inset on the bottom-right-hand corner shows the names of the countries and sea areas within the investigation area.

**Figure 2.** Model domains used for the nesting of the CCLM simulations.

**Figure 3.** Surface weather chart of mean sea level pressure (hPa), fronts and surface troughs at (a) 1200 UTC and (b) 1800 UTC on 3 January 2014 (source: UK Met Office). The surface trough associated with the development of the derecho-producing mesoscale convective system is denoted by the orange line.


**Figure 4.** ERA5 reanalysis of the synoptic-scale conditions at 1500 UTC and 1800 UTC on 3 January 2014. (a)-(b) Mean sea level pressure (hPa; black lines) and hourly pressure tendency (hPa h$^{-1}$; shaded), (c)-(d) 500 hPa wind speed (m s$^{-1}$; contour lines starting at 25 m s$^{-1}$) and divergence (10$^{-5}$ s$^{-1}$; shaded), (e)-(f) 500 hPa geopotential height (gpm; black lines) and 700 hPa upward motion (Pa s$^{-1}$; shaded). The dashed lines in (a) and (b) denotes the surface pressure trough. The bold black
lines in (c)-(f) denote the location of the cross sections shown in Fig. 5.

**Figure 5.** West-east-orientated cross sections depicting temperature (K; black lines), horizontal divergence/convergence (10$^{-5}$ s$^{-1}$; shaded) and vertical velocity (Pa s$^{-1}$; black arrows) at (a) 1500 UTC (cf. Fig 4e) and at (b) 1800 UTC (cf. Fig. 4f).

Figure 6. ERA5 reanalysis of (a)-(b) 950 hPa specific humidity (g kg$^{-1}$), (c)-(d) 900-650-hPa lapse rate (K km$^{-1}$), (e)-(f) 0-6
km bulk shear (m s$^{-1}$; shaded) and (e)-(f) 0-3 km bulk shear larger than 15 m s$^{-1}$ (hatched areas) at (a),(c),(e) 1200 UTC and (b),(d),(f) 1800 UTC on 3 January 2014. The white dot in (a),(c) and (e) denotes the location of the sounding shown in Fig. 7. The dashed white line indicates the position of the surface trough according to the UK Met Office surface analysis shown in Fig. 3.






**Figure 7.** Skew T-log p diagram of upper-air observations from Larkhill (England) at 1200 UTC on 3 January 2014. The thick red (blue) line represent temperature (dew point) values in °C. The box in the upper right corner shows the values for 50-hPa mixed-layer and most-unstable CAPE/CIN, bulk shear and precipitable water (PWAT). The insets on the right-hand side show the vertical distribution of the horizontal wind (kn; wind barbs) and of the most-unstable CAPE/CIN (J kg$^{-1}$; brown line).

**Figure 8.** Event probability forecast valid for 0000 UTC on 4 January 2014 by the 0000 UTC run of (a),(c) ECMWF-EPS and (b),(d) COSMO-LEPS on 3 January 2014 in terms of (a)-(b) maximum 10 m AGL wind gusts exceeding 20 m s$^{-1}$ within 24 hours and (c)-(d) maximum 10 m AGL wind gusts exceeding 25 m s$^{-1}$ within 24 hours.

**Figure 9.** Results from (a)-(b) CCLM-ERA5, (c)-(d) CCLM-ERAI and (e)-(f) CCLM-ECAN at 1600 UTC. (a),(c),(e) 1-hourly mean sea level pressure (MSLP) tendency (hPa h$^{-1}$; shaded) and 950 hPa convergence smaller than -5 x 10$^{-5}$ s$^{-1}$ (hatched areas) from the 7-km simulation. (b),(d),(f) Column maximum reflectivity (dBZ; shaded) and 50-hPa mixed-layer CAPE above 50 J kg$^{-1}$ (hatched areas) from the 2.8-km simulation.

**Figure 10.** Results from CCLM-ERA5 7-km simulation with (a),(c) hourly and (b) 6-hourly lateral boundary conditions (LBCs). (a)-(b) 1-hourly mean sea level pressure (MSLP) tendency (hPa h$^{-1}$; shaded) at 1400 UTC, (c) MSLP (hPa; shaded) at 0900 UTC and (d) MSLP difference (hPa; shaded) at 0900 UTC between simulations with hourly and 6-hourly LBCs. The black outlined box in (c) highlights the surface pressure trough, which entered the model domain from the west.

**Figure 11.** Results from CCLM-ERA5 at (a)-(b) 1400 UTC, (c) 1600 UTC and (d) 1900 UTC. (a) 1-hourly mean sea level pressure (MSLP) tendency (hPa h$^{-1}$; shaded) and 950 hPa convergence smaller than -5 x 10$^{-5}$ s$^{-1}$ (hatched areas) from the 7-km simulation. (b)-(d) Column maximum reflectivity (dBZ; shaded) and 50-hPa mixed-layer CAPE above 50 J kg$^{-1}$ (hatched areas) from the 1.1-km simulation. The yellow dashed lines in (a) denote the convection-initiating boundaries.

**Figure 12.** Comparison of the observed and modeled reflectivity at approximately 1.5 km altitude at (a)-(b) 1500 UTC and at (c)-(d) 1700 UTC. (a) RMIB radar reflectivity composite (dBZ), (c) KNMI radar reflectivity composite (dBZ) and (b),(d) reflectivity from the CCLM-ERA5 1.1-km simulation.

**Figure 13.** CCLM-ERA5 2.8-km and 1.1-km simulations of (a)-(b) 10 m AGL maximum wind gusts (m s$^{-1}$) and (c)-(d) maximum mean sea level pressure (MSLP) gradient (Pa km$^{-1}$) between 1400 and 2200 UTC.



540    **Table 1.** Specifications about the physical parameterisations used in the different CCLM domains.

| Domain | D0 (ERAI) | D1 (ERAI, ECAN, ERA5) | D2 (ERAI, ECAN, ERA5) | D3 (ERA5) |
|---|---|---|---|---|
| Horizontal grid spacing | 0.22° (Δx ≈ 25 km) | 0.0625° (Δx ≈ 7 km) | 0.025° (Δx ≈ 2.8 km) | 0.01° (Δx ≈ 1.1 km) |
| No. of vertical layers | 40 | 50 | 60 | 90 |
| Convective parameterisation | Tiedtke (1989) | | Only shallow convection after Tiedtke (1989) | |
| Cloud microphysics | Two-Category Ice Scheme (Doms et al., 2011) | | Three-Category Ice or Graupel Scheme (Reinhardt and Seifert, 2006) | |
| Radiation | Ritter and Geleyn (1992); Rockel et al. (1991) | | | |
| Soil model | Multi-layer soil model (TERRA-ML) after Jacobsen and Heise (1982) | | | |
| Planetary boundary layer turbulence | Baldauf et al. (2011); Mellor and Yamada (1982) | | | |




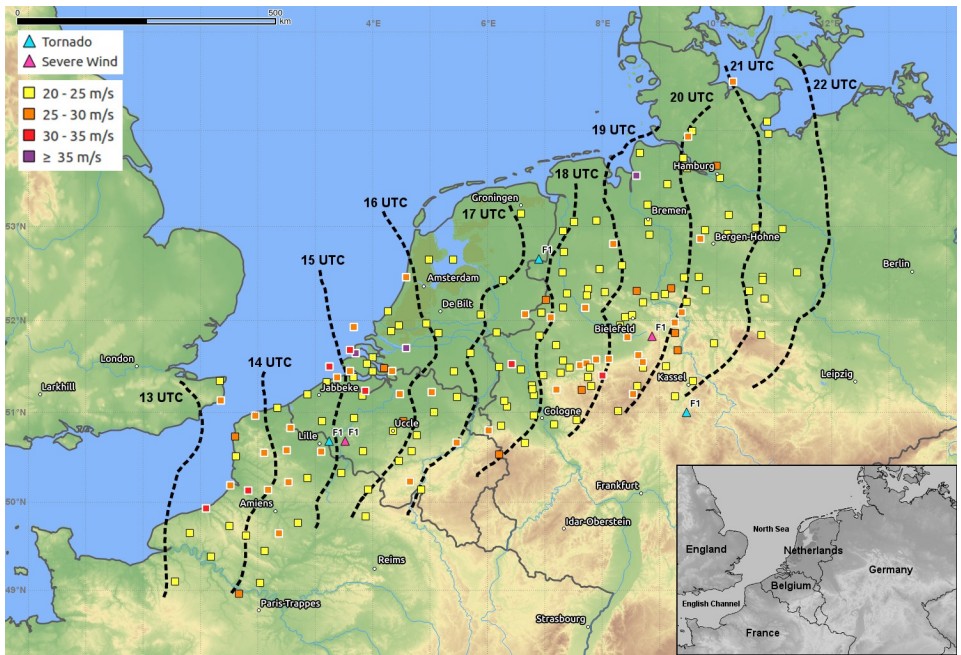

**Figure 1.** The radar-observed position of the leading edge of the derecho-producing mesoscale convective system at hourly intervals between 1300 and 2200 UTC on 3 January 2014 is shown by the dashed black lines. The observed maximum wind gusts (m s$^{-1}$) are denoted by the small colored squares (see legend). The white edged squares indicate gusts stronger than 25.7 m s$^{-1}$. Tornadic and nontornadic wind damage locations are marked by the small blue and magenta triangles, respectively (see legend). The inset on the bottom-right-hand corner shows the names of the countries and sea areas within the investigation area.



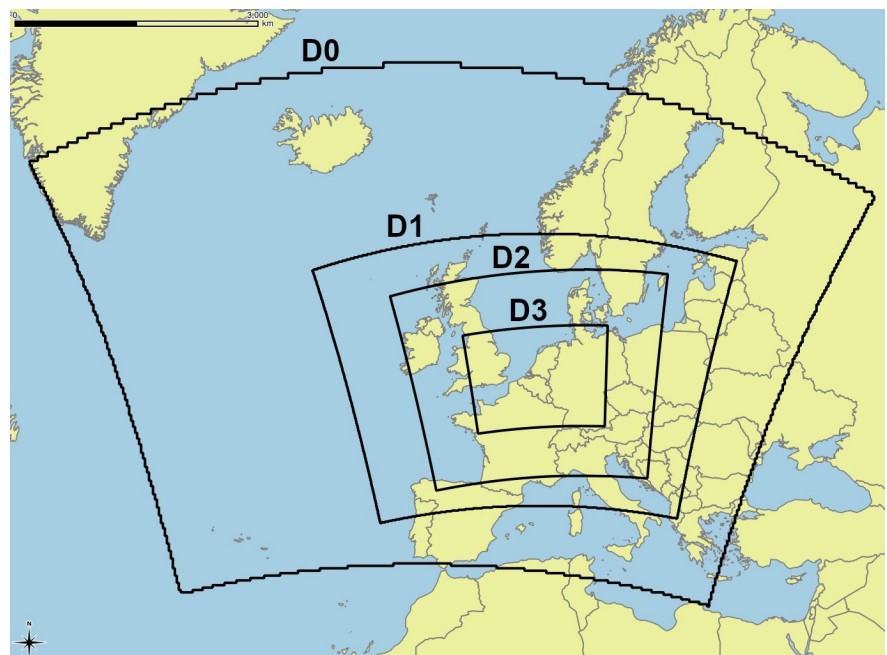

**Figure 2.** Model domains used for the nesting of the CCLM simulations.




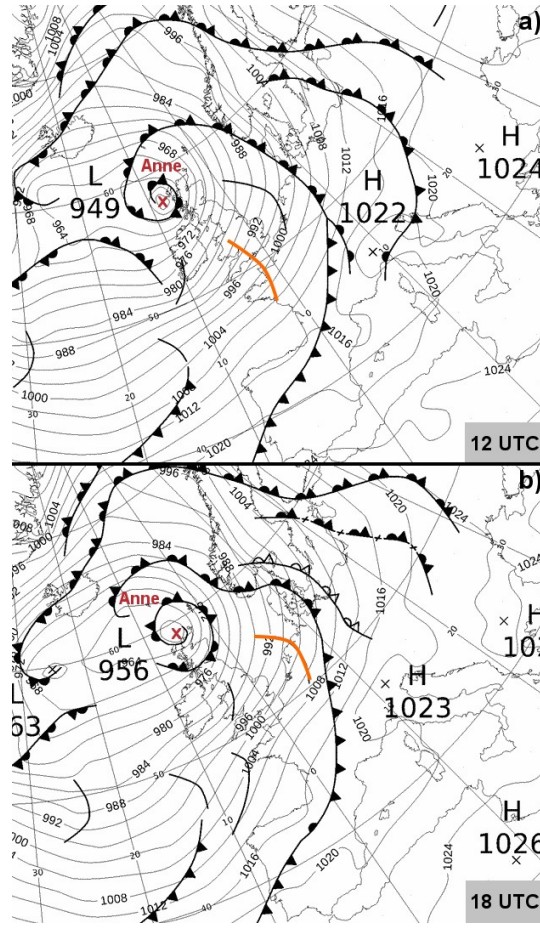

**Figure 3.** Surface weather chart of mean sea level pressure (hPa), fronts and surface troughs at (a) 1200 UTC and (b) 1800 UTC on 3 January 2014 (source: UK Met Office). The surface trough associated with the development of the derecho-producing mesoscale convective system is denoted by the orange line.





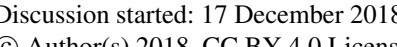

**Figure 4.** ERA5 reanalysis of the synoptic-scale conditions at 1500 UTC and 1800 UTC on 3 January 2014. (a)-(b) Mean
sea level pressure (hPa; black lines) and hourly pressure tendency (hPa h$^{-1}$; shaded), (c)-(d) 500 hPa wind speed (m s$^{-1}$;
contour lines starting at 25 m s$^{-1}$) and divergence (10$^{-5}$ s$^{-1}$; shaded), (e)-(f) 500 hPa geopotential height (gpm; black lines) and
700 hPa upward motion (Pa s$^{-1}$; shaded). The dashed lines in (a) and (b) denotes the surface pressure trough. The bold black
lines in (c)-(f) denote the location of the cross sections shown in Fig. 5.



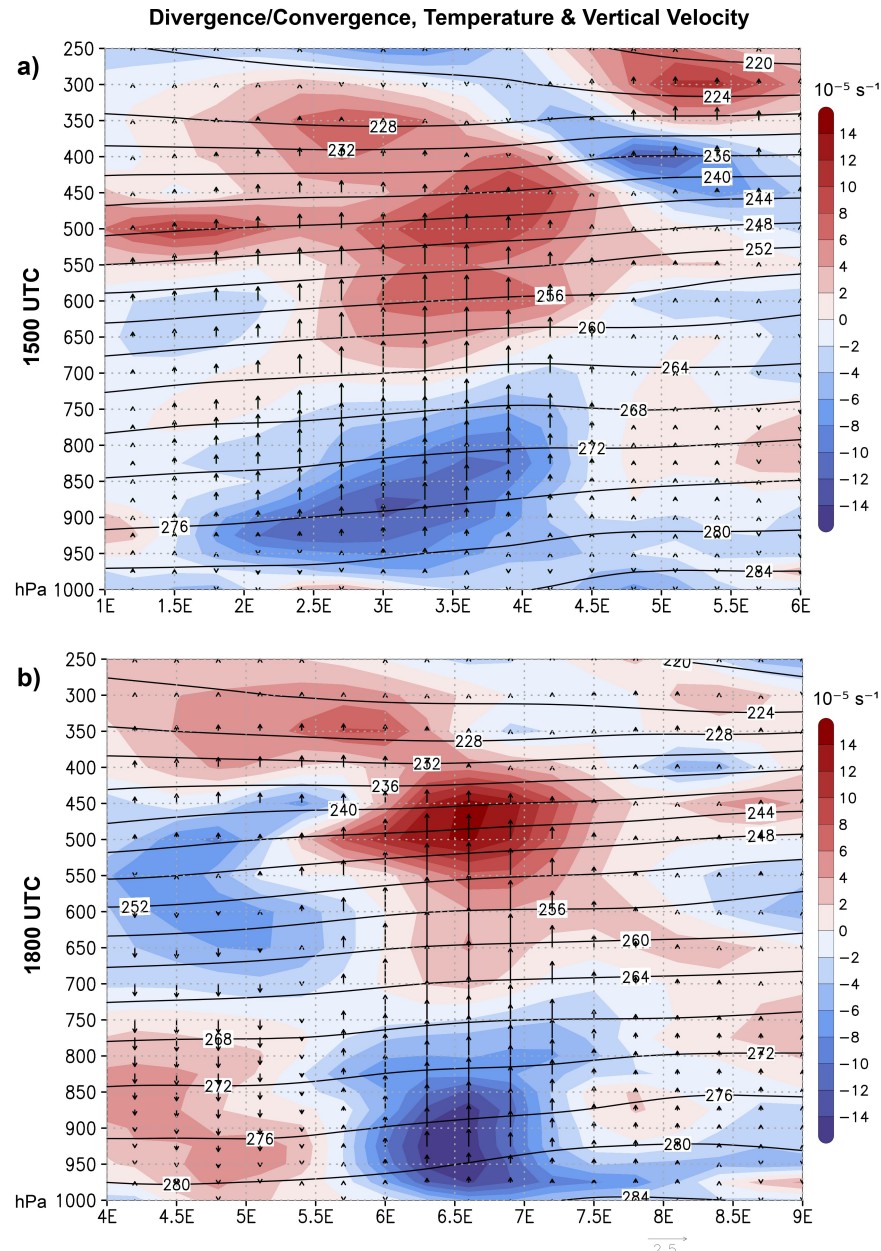

**Figure 5.** West-east-orientated cross sections depicting temperature (K; black lines), horizontal divergence/convergence (10^-5 s^-1; shaded) and vertical velocity (Pa s^-1; black arrows) at (a) 1500 UTC (cf. Fig 4e) and at (b) 1800 UTC (cf. Fig. 4f).

**Figure 6.** ERA5 reanalysis of (a),(b) 950 hPa specific humidity (g kg$^{-1}$), (c),(d) 900-650-hPa lapse rate (K km$^{-1}$), (e),(f) 0-6 km bulk shear (m s$^{-1}$; shaded) and (e),(f) 0-3 km bulk shear larger than 15 m s$^{-1}$ (hatched areas) at (a),(c),(e) 1200 UTC and (b),(d),(f) 1800 UTC on 3 January 2014. The white dot in (a),(c) and (e) denotes the location of the sounding shown in Fig. 7. The dashed white line indicates the position of the surface trough according to the UK Met Office surface analysis shown in Fig. 3.



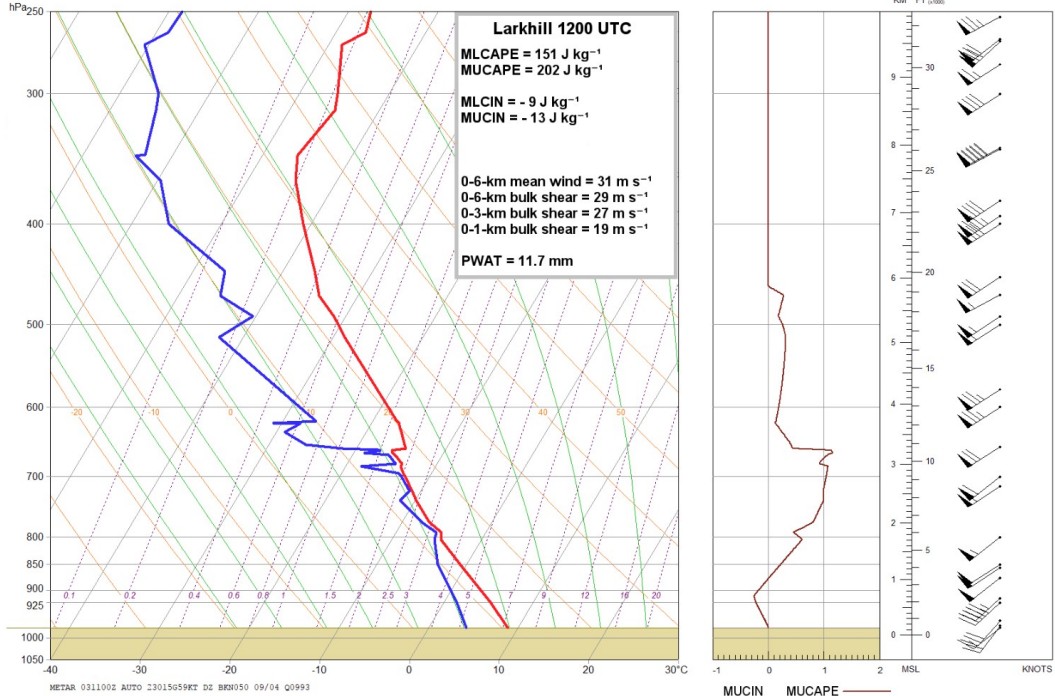

**Figure 7.** Skew T-log p diagram of upper-air observations from Larkhill (England) at 1200 UTC on 3 January 2014. The thick red (blue) line represent temperature (dew point) values in °C. The box in the upper right corner shows the values for 50-hPa mixed-layer and most-unstable CAPE/CIN, bulk shear and precipitable water (PWAT). The insets on the right-hand side show the vertical distribution of the horizontal wind (kn; wind barbs) and of the most-unstable CAPE/CIN (J kg⁻¹; brown line).





**Figure 8.** Event probability forecast valid for 0000 UTC on 4 January 2014 by the 0000 UTC run of (a),(c) ECMWF-EPS and (b),(d) COSMO-LEPS on 3 January 2014 in terms of (a)-(b) maximum 10 m AGL wind gusts exceeding 20 m s$^{-1}$ within 24 hours and (c)-(d) maximum 10 m AGL wind gusts exceeding 25 m s$^{-1}$ within 24 hours.





**Figure 9.** Results from (a)-(b) CCLM-ERA5, (c)-(d) CCLM-ERAI and (e)-(f) CCLM-ECAN at 1600 UTC. (a),(c),(e) 1-hourly mean sea level pressure (MSLP) tendency (hPa h$^{-1}$; shaded) and 950 hPa convergence smaller than -5 x 10$^{-5}$ s$^{-1}$ (hatched areas) from the 7-km simulation. (b),(d),(f) Column maximum reflectivity (dBZ; shaded) and 50-hPa mixed-layer CAPE above 50 J kg$^{-1}$ (hatched areas) from the 2.8-km simulation.



**Figure 10.** Results from CCLM-ERA5 7-km simulation with (a),(c) hourly and (b) 6-hourly lateral boundary conditions (LBCs). (a)-(b) 1-hourly mean sea level pressure (MSLP) tendency (hPa h⁻¹; shaded) at 1400 UTC, (c) MSLP (hPa; shaded) at 0900 UTC and (d) MSLP difference (hPa; shaded) at 0900 UTC between simulations with hourly and 6-hourly LBCs. The black outlined box in (c) highlights the surface pressure trough, which entered the model domain from the west.







**Figure 11.** Results from CCLM-ERA5 at (a)-(b) 1400 UTC, (c) 1600 UTC and (d) 1900 UTC. (a) 1-hourly mean sea level pressure (MSLP) tendency (hPa h$^{-1}$; shaded) and 950 hPa convergence smaller than -5 x 10$^{-5}$ s$^{-1}$ (hatched areas) from the 7-km simulation. (b)-(d) Column maximum reflectivity (dBZ; shaded) and 50-hPa mixed-layer CAPE above 50 J kg$^{-1}$ (hatched areas) from the 1.1-km simulation. The yellow dashed lines in (a) denote the convection-initiating boundaries.





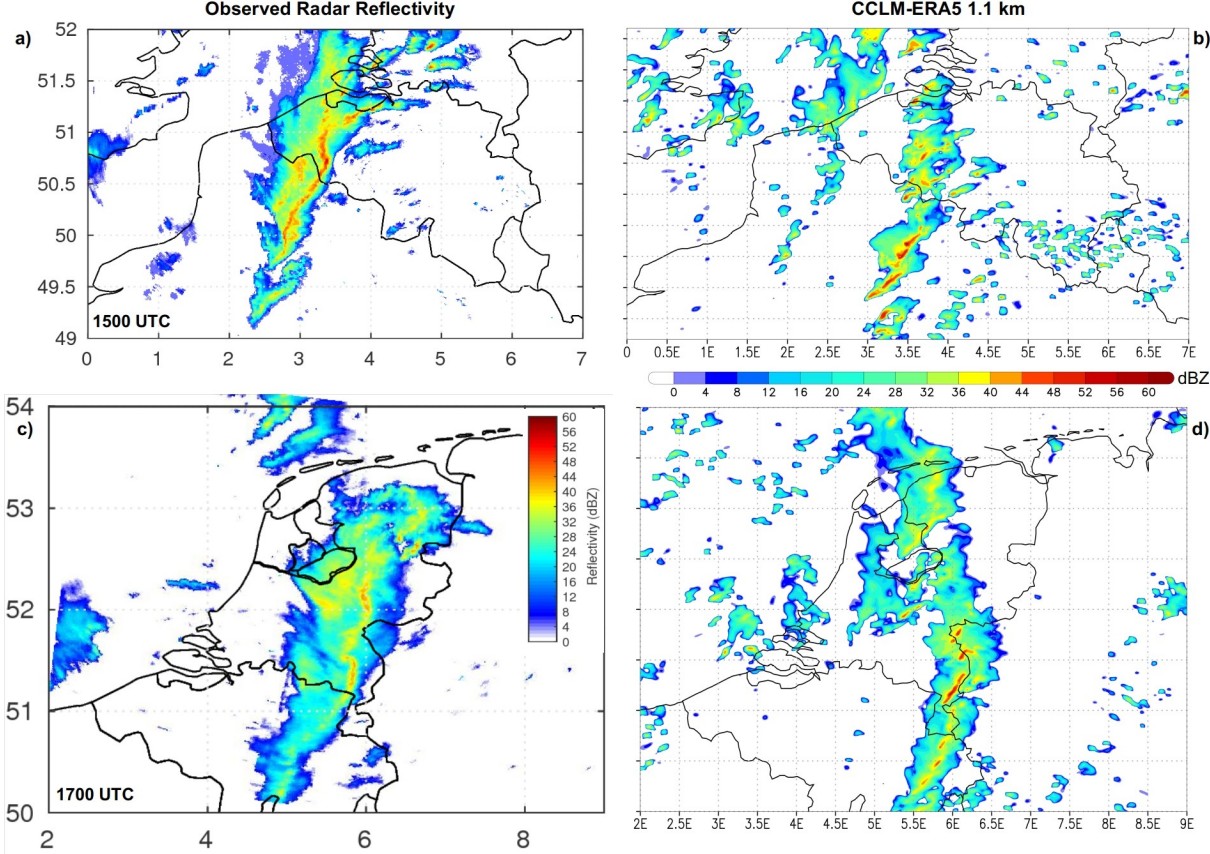

**Figure 12.** Comparison of the observed and modeled reflectivity at approximately 1.5 km altitude at (a)-(b) 1500 UTC and at (c)-(d) 1700 UTC. (a) RMIB radar reflectivity composite (dBZ), (c) KNMI radar reflectivity composite (dBZ) and (b),(d) reflectivity from the CCLM-ERA5 1.1-km simulation.





**Figure 13.** CCLM-ERA5 2.8-km and 1.1-km simulations of (a)-(b) 10 m AGL maximum wind gusts (m s$^{-1}$) and (c)-(d) maximum mean sea level pressure (MSLP) gradient (Pa km$^{-1}$) between 1400 and 2200 UTC.