# Peer review of "Synoptic-scale conditions and convection-resolving hindcast experiments of a cold-season derecho on 3 January 2014 in Western Europe"

_Natural Hazards and Earth System Sciences, 2018_

## Referee Comment (RC1) · Leutwyler (Referee) · 3 Jan 2019

In the manuscript Mathias et al. present a case study of a linearly-organized MCS and its representation in convection-resolving and convection-parameterizing simulations. Apparently, the case was unusual because a derecho developed in a postfrontal airmass of an extratropical cyclone. They start by describing the case in detail based on ERA5 reanalysis and subsequently perform a set of simulations to assess model and configuration sensitivities. The three main findings are as following: (1) Explicitly resolving convection leads to added value when forecasting this derecho. (2) Further

added value has been found when refining grid spacing from 2.8 km to 1.1 km. (3) In limited-area models, lateral boundary conditions need to be updated hourly to represent the case correctly.

The manuscript is novel, interesting and fits well into the scope of NHESS, but could benefit from addressing style, syntax, and conciseness.

Major Comments

1. I think the claim that this case was "not well anticipated" is exaggerated. While similar statements are repeated throughout the text, they are not well justified since storm-force wind was actually mentioned in the weather reports (P2L36 – P2L43). Furthermore, since some features are represented in ERA5, the identified deficiencies in the NWP simulations (Figure 8) point towards an issue in the observational analysis at the time (as mentioned in the summary).

I suggest to remove this entire discussion and shift the motivation towards added value of explicit convection for these events. It is interesting enough, as so far most of the discussion about resolving convection explicitly has been about diurnal convection (see Prein et al., 2015) or the tropics. Further motivation, specific to the NHESS audience, could be based on the discussions around global early warning systems (see Copernicus systems) and the question whether convection-resolving resolution is needed, or if resources should be invested in more ensemble members.

2. I would choose to configure the simulation configurations as identically as possible. Apart from the product to derive the initial and lateral boundary conditions, you have chosen to vary the number of intermediate nests, the init time, the microphysics scheme and the number of vertical layers. That makes it hard to pinpoint the observed differences to specific changes in configurations. For instance, at P8L256 you can't distinguish between differences in vertical and horizontal resolution. I am not sure if the vertical resolution is the key issue.

P4L120- P5L135: I am a bit confused by the rather complicated setup chosen (maybe add a Gantt-chart-type figure outlining the init and update time?).

P4L126: Delete: "The aforementioned temporal setup is used to permit a most realistic simulation of the derecho and allow a direct comparison of the simulated data with the observational data."

3. I find the mix of panels showing different fields from different simulation resolutions in the same figure a bit confusing. I would switch between two modes. (i) When comparing model resolution I would always show the same fields for 7km, 2.8 km, and 1.1 km. (ii) When comparing between driving datasets, I would show the same panels for 7 km and 2.8 km.

Also, I would discuss the validation in Fig. 12 before the sensitivity studies.

3. P7L224-P8L228: "would have been the key factors to successfully forecast this cold-season storm." Either I am confused, or you jump to conclusions too fast. You mix the influence of LBC update frequency and forecast skill in ECAN, which is a global simulation. I guess you arrived at your conclusion because Figure 9e looks a bit like Figure 10b, right?

Would it be possible to show time series of the driving fields (ERAI, ERA5, ECAN)? Maybe in the supplement?

Minor Comments

1. P1L26: There are earlier references introducing the concept of extra-tropical cyclones than Ludwig et al. (2015) and Gatzen (2018).

2. Section 3: Provide a concise summary of the criteria needed for an event to classify as a derecho and how they apply here. Now, this discussion is scattered throughout the manuscript.

3. P7L194-P7L200: Maybe outline if the environment was at least predicted correctly?

Explain why the underestimation of wind speed cannot be attributed to the wind gust parametrization.

4. Figure 5: Why do you show fields from two different resolutions: (left) 7km and (right) 2.8 km (see above)?

5. Figure 12: Add 2.8 km reflectivity?

Technical Changes

1. P3L70 – P3L86 These paragraphs need work. Your co-authors should be able to tell you how to make it more readable.

2. P2L44 – P3L66: Reading a list of papers (a reader probably doesn't know) without much context does not motivate to continue reading. Put the literature in context, explain where the gap in research is and why you think it is interesting.

3. P5L37 - P5L153: Although I enjoyed reading the detailed description of the evolution of cyclone Anne, there might be potential to shorten the text here.

4. P5L154-P6L187 While it is certainly a good idea to spend a bit of time explaining convection precursors to NHESS readers, there might be some potential to shorten this part too.

5. P7L217 – P7220 From a technical perspective this is an interesting result you may want to highlight more. There is an ongoing discussion about resolution jump vs. LBC update frequency for limited area modeling. Also, specify the employed upper boundary condition (Relaxation or w=0?) in Section 2.

Figures 2 -7: In the beginning, 5 figures are shown to set the stage for the main ideas following. Are all of them needed? Along with the shortening potential in (3) and (4). There might be potential to remove some of the panels or move them to the supplement.

P1L11: trough favored → trough, and was favored

P1L12: conditions were → environment was

P1L14: You need to mention that these are limited-area simulations

P1L13: latent instability. Maybe use the more common term conditional instability (also rest of text)?

P1L15: datasets → reanalysis datasets

P1L15: I would write "initial and lateral boundary conditions derived from ERA5"

P1L17: (i) convection-resolving scale → convection-resolving resolution (ii) At P1L14 you use the term convection-permitting. I would try and use just one of the two. We usually use convection-resolving, since the

P1L18: This case study is testimony to the usefulness of ensembles of convection-resolving simulations to . . .

P1L21: affect ← wrong word

P1L22: The style of the manuscript is unnecessarily cautious (hedging), which is legitimate to protect your claims, however, in most cases, it is actually not needed. For example: "In some cases, MCSs can exhibit". There is no need to add another can here. Check in the entire manuscript if vague language is really needed.

P1L26: (i) remove: "which is" (ii) linearly organized → linearly-organized (also address hyphenation mistakes in rest of text)

P1L28: remove: "region"

P2L45: It is pointed out ← rewrite

P2L50: AGL ← define

P3L83: grid interval → grid spacing

P4L95: realised → conducted

P4L95: taken → derived

P4L110: Citing Baldauf et al (2011) may be warranted

P5L138: intensive → deep

P5L149: analysed → diagnosed

P7L206: "displaced" Wrong word, since in this Section, we don't know (yet) the true location.

P8L231: the highest-resolution run → the simulation with 1 km grid spacing

P8L235: "convection-initiating boundaries". Maybe choose a different term as it can be confused with the lateral boundaries, which you also discuss. Maybe "lifting mechanism"?

P10L308-L310: Maybe mention that at least DWD and MCH employ such systems these days.

Table 1: Specifications about the physical parameterisations used in the different CCLM domains. → Simulation configurations

Figure 2 caption: I usually use the term "computational domain" (also in rest of text)

Figure 4: diagnosed 700 hPa upward motion.

---

## Referee Comment (RC2) · Anonymous Referee #2 · 13 Feb 2019

This paper presents a series of high-resolution simulations of a high-impact winter severe weather event in Europe. This event was poorly represented in the operational FC and the paper addresses the question of why this was the case. The results very nicely shows that very high temporal resolution boundary condition input is required to capture the event with high resolution simulations. The main findings are clearly communicated and well documented and I have only minor questions and requests for changes.

L22 endure last L23 for a broader readership consider to define the term derecho L24

might may L29 Please define straight-line wind damage L33/34 under suspicion of being produced by were likely affected by tornados L43 It is unclear what is meant by "these specific characteristics" L46 Please define strongly forced L51 with regard to for L101 the statement "which in the end leads to two additional distinct input data sets" is unclear L133 What do you mean by the statement "while the ECAN boundary conditions remain unchanged? L144 discontinuity of what? L153 I am more familiar with the term anticyclonically tilted L162ff Why was the event not recognized by the forecasters if all the ingredients were so clearly present? L185 The high shear values are of course related to the fact that Kyrill was one of the strongest storms in this area in the last decades, maybe add a comment. L194ff: Could you in addition to the surface wind signature briefly comment on how the env. Conditions for convection (shear, stability etc.) were represented in the fc? L209 Do you know why the trough was missing in the simulations? Were dry or moist dynamics responsible for this fc failure? L225 Related to the previous point, how exactly did the trough form? L240 Please define low-end CAPE L252 Could you add the observations to figure 13, going back and forth to figure 1 makes the comparison quite cumbersome. L265 Please define linear upscale growth L267 Can you really call a trough a boundary?

Figures: The line labels are in most figures difficult to read and I recommend increasing their size Figure 1: can you highlight the location of the Larkhill sounding more prominently? Figure 4f: Do you have in indication if the upward motion is mainly in response to diabatic heating or due to qg forcing? Figure 7: I am not familiar with this graphical representation of the MU cape. Are the unit values really around 1 J/kg? How do the values add up to 202 J/kg?

---

## Author Comment (AC1) · 25 Feb 2019

This paper presents a series of high-resolution simulations of a high-impact winter severe weather event in Europe. This event was poorly represented in the operational FC and the paper addresses the question of why this was the case. The results very nicely shows that very high temporal resolution boundary condition input is required to capture the event with high resolution simulations. The main findings are clearly communicated and well documented and I have only minor questions and requests for changes.

A: We would like to thank the reviewer for his/her time spent on the manuscript and his/her thoughtful comments that helped to improve the manuscript. Point-to-point responses to each comment can be found below (marked in red). We have included a careful justification for those points where we did not fully follow the suggestions by the reviewer.

L22 endure last

A: We replaced "endure" with "last".

L23 for a broader readership consider to define the term derecho

A: We have now included a complete definition of the term "derecho" in the introduction.

L24 might may

A: We replaced "might" with "may".

L29. Please define straight-line wind damage

A: We replaced "straight-line wind damage" with "non-tornadic wind damage".

L33/34 under suspicion of being produced by were likely affected by tornados.

A: We deleted that part of the sentence.

L43 It is unclear what is meant by "these specific characteristics"

A: We replaced "These" with "All the above-mentioned" to make this statement clearer.

L46 Please define strongly forced

A: "Strongly forced" is a term which is generally used in the literature to describe synoptic situations with strong large-scale/QG forcing for ascent.

L51 with regard to for

A: We replaced "with regard to" with "for".

L101 the statement "which in the end leads to two additional distinct input datasets" is unclear

A: We removed that statement.

L133 What do you mean by the statement "while the ECAN boundary conditions remain unchanged"?

A: It means that the wind and moisture fields of the ECAN boundary conditions were not changed, in contrast to the wind and moisture fields in the initial conditions.

L144 discontinuity of what?

A: We deleted the term "discontinuity" to avoid confusion.

L153 I am more familiar with the term anticyclonically tilted

A: "Negatively tilted" means that the trough axis has a NW-SE-orientation.

L162ff Why was the event not recognized by the forecasters if all the ingredients were so clearly present?

A: Two important ingredients - the lifting mechanism (surface trough) and the latent instability over Benelux - were not fully anticipated in the forecasts of the event.

L185 The high shear values are of course related to the fact that Kyrill was one of the strongest storms in this area in the last decades, maybe add a comment.

A: We added "in a highly baroclinic environment".

L194ff: Could you in addition to the surface wind signature briefly comment on how the env. Conditions for convection (shear, stability etc.) were represented in the fc?

A: We added a sentence mentioning that the ECMWF-EPS underestimated the latent instability over Benelux (see figure below).

[Figure]

**Figure R1.** Probability of CAPE being larger than 50 J/kg at 1800 UTC on 3 January 2014.

L209 Do you know why the trough was missing in the simulations? Were dry or moist dynamics responsible for this fcfailure? L225 Related to the previous point, how exactly did the trough form?

A: The formation of the surface trough associated with the DMCS was likely linked to another surface trough moving to Ireland. We have shown that with the different update frequencies of LBCs in the CCLM-ERA5 simulations.

L240 Please define low-end CAPE

A: It is defined as CAPE < 150 J/kg, which is indicated in the brackets.

L252 Could you add the observations to figure 13, going back and forth to figure 1 makes the comparison quite cumbersome.

A: Our intention is not that the reader should compare every point in situ observations with the model fields. We only wanted to provide a general comparison of the overall gust strength, and hence we chose to use the same colorbar.

L265 Please define linear upscale growth

A: Formation of a more or less closed convective line based on initially scattered convective cells.

L267 Can you really call a trough a boundary?

A: We changed the term to "convection-initiating convergence zone".

Figures: The line labels are in most figures difficult to read and I recommend increasing their size

A: We increased the size of the line labels in Figure 4 and removed the contour lines in Figure 6. We removed Figure 5 entirely. Overall, we enhanced the readability of our figures.

Figure 1: can you highlight the location of the Larkhill sounding more prominently?

A: Larkhill is now highlighted more prominently with a dark blue dot.

Figure 4f: Do you have in indication if the upward motion is mainly in response to diabatic heating or due to qg forcing?

A: Mainly in response to QG forcing, as it was associated with an amplifying/digging upper-level trough.

Figure 7: I am not familiar with this graphical representation of the MU cape. Are the unit values really around 1 J/kg? How do the values add up to 202 J/kg?

A: We changed the graphical representation of MUCAPE and MUCIN (see Fig. R2). It should be clearer now.

[Figure]

**Figure R2.** Updated Fig. 6 (old Fig.7). MUCAPE (MUCIN) is denoted by the red (blue) area between the temperature profile and the parcel ascent curve.

---

## Author Comment (AC2) · 25 Feb 2019

In the manuscript Mathias et al. present a case study of a linearly-organized MCS and its representation in convection-resolving and convection-parameterizing simulations. Apparently, the case was unusual because a derecho developed in a postfrontal airmass of an extratropical cyclone. They start by describing the case in detail based on ERA5 reanalysis and subsequently perform a set of simulations to assess model and configuration sensitivities. The three main findings are as following: (1) Explicitly resolving convection leads to added value when forecasting this derecho. (2) Further added value has been found when refining grid spacing from 2.8 km to 1.1 km. (3) In limited-area models, lateral boundary conditions need to be updated hourly to represent the case correctly.

The manuscript is novel, interesting and fits well into the scope of NHESS, but could benefit from addressing style, syntax, and conciseness.

A: We would like to thank the reviewer for his time spent on the manuscript and his thoughtful comments that helped to improve the manuscript. Point-to-point responses to each comment can be found below (marked in red). We have included a careful justification for those points where we did not fully follow the suggestions by the reviewer.

**Major Comments**

1. I think the claim that this case was "not well anticipated" is exaggerated. While similar statements are repeated throughout the text, they are not well justified since storm-force wind was actually mentioned in the weather reports (P2L36 – P2L43). Furthermore, since some features are represented in ERA5, the identified deficiencies in the NWP simulations (Figure 8) point towards an issue in the observational analysis at the time (as mentioned in the summary).

A: It is true that the potential for storm-force/severe gusts was mentioned in the official weather reports, but only in association with isolated thundery showers. While ESTOFEX did hint at the possibility of a convective line ("A compact line of deeper convection could also cause a swath of severe wind gusts"), their LVL1 area did not cover the area affected by the derecho, as we already mentioned in the introduction. In particular, a meteorologist from the Belgian weather service RMIB even stated in a media interview that they anticipated gusts up to 25 m s$^{-1}$ in western Belgium within the synoptic-scale flow (cf. EPS probabilities), but not in association with deep moist convection (source: https://www.levif.be/actualite/belgique/des-rafales-de-vent-jusqu-a-90-km-h-prevues-mais-pas-combinees-a-des-orages/article-normal-15959.html?cookie_check=1550580823).

Therefore, we concluded that the 3 January 2014 long-lived convective windstorm was not well anticipated by the weather forecasting centers.

I suggest to remove this entire discussion and shift the motivation towards added value of explicit convection for these events. It is interesting enough, as so far most of the discussion about resolving convection explicitly has been about diurnal convection (see Prein et al., 2015) or the tropics. Further motivation, specific to the NHESS audience, could be based on the discussions around global early warning systems (see Copernicus systems) and the question whether convection-resolving resolution is needed, or if resources should be invested in more ensemble members.

A: We thank the reviewer for this suggestion. While we do understand your point of view, we would like to keep this discussion on predictability. We think that this section is very important as the magnitude of the event was not well anticipated in the forecasts, which we tried to describe in more detail. The question whether convection-resolving resolution or more ensemble members are needed in general cannot be generally answered within the scope of this case study. For this specific event, the 1.1 km simulation is the most realistic depiction of the event and shows some added value regarding simulated radar reflectivity and gust strength compared to the 2.8 km simulation. Still, the consideration of a bigger ensemble and different update times of boundary conditions finally leads to a realistic representation of the event.

2. I would choose to configure the simulation configurations as identically as possible. Apart from the product to derive the initial and lateral boundary conditions, you have chosen to vary the number of intermediate nests, the init time, the microphysics scheme and the number of vertical layers. That makes it hard to pinpoint the observed differences to specific changes in configurations. For instance, at P8L256 you can't distinguish between differences in vertical and horizontal resolution. I am not sure if the vertical resolution is the key issue.

P4L120- P5L135: I am a bit confused by the rather complicated setup chosen (maybe add a Gantt-chart-type figure outlining the init and update time?).

A: We wanted to achieve an identical start time for all the high-resolution (2.8 km) runs (i.e., 1200 UTC). To reach this goal, different nesting steps were required because the various ILBCs have different spatial resolutions. Moreover, a simultaneous increase of the vertical and horizontal resolution is a common method implemented by national weather services (e.g., DWD, Météo-France) to obtain a numerically stable convection-permitting/resolving NWP. We added a Gantt chart for a better overview of the different CCLM configurations and initialisation times to Figure 2. Additionally, we modified the corresponding paragraph, and hope that the rather complicated set up is now understandable

[Figure]

**Figure R1.** (a) Computational domains used for the nesting of the CCLM simulations and (b) Gantt chart overview of the different CCLM configurations and initialisation times. (updated Fig. 2)

P4L126: Delete: "The aforementioned temporal setup is used to permit a most realistic simulation of the derecho and allow a direct comparison of the simulated data with the observational data."

The sentence has been deleted as requested by the reviewer; nevertheless, we kept a hint that a series of simulations with different starting times for the different ILBC's have been performed to select the runs for analysis with the closest agreement to the observed event.

3. I find the mix of panels showing different fields from different simulation resolutions in the same figure a bit confusing. I would switch between two modes. (i) When comparing model resolution I would always show the same fields for 7km, 2.8 km, and 1.1 km. (ii) When comparing between driving datasets, I would show the same panels for 7 km and 2.8 km.

A: We chose the 7 km simulation to visualize the pressure tendency and low-level convergence because of its smoother fields, which are easier to interpret. Hence, we would like to retain the current panel setup.

Also, I would discuss the validation in Fig. 12 before the sensitivity studies.

A: We thank the reviewer for this suggestion. However, we would like to stick with our current logical structure of the manuscript. This (updated) Figure shows nicely the better matching of the simulated 1.1 km with the observed reflectivity compared to the simulated 2.8 km reflectivity pattern.

[Figure]

**Figure R2.** Updated Fig. 11 (old Fig. 12).

4. P7L224-P8L228: "would have been the key factors to successfully forecast this cold-season storm." Either I am confused, or you jump to conclusions too fast. You mix the influence of LBC update frequency and forecast skill in ECAN, which is a global simulation. I guess you arrived at your conclusion because Figure 9e looks a bit like Figure 10b, right?

A: We arrived at this conclusion because the surface pressure trough associated with the derecho only existed in the ERA5-driven CCLM simulation with hourly LBCs. This trough did not form in CCLM-ERA5 with 6-hourly LBCs. As a result, the MSLP tendency field of CCLM-ECAN looks similar to CCLM-ERA5 with 6-hourly LBCs at 1400 UTC.

Would it be possible to show time series of the driving fields (ERAI, ERA5, ECAN)? Maybe in the supplement?

A: We thank the reviewer for this suggestion, but we think that this aspect is too technical for a peer review manuscript.

**Minor Comments**

1. P1L26: There are earlier references introducing the concept of extra-tropical cyclones than Ludwig et al. (2015) and Gatzen (2018).

A: The two references are specifically related to winter derechos along cold fronts and not to the general concept of ETCs.

2. Section 3: Provide a concise summary of the criteria needed for an event to classify as a derecho and how they apply here. Now, this discussion is scattered throughout the manuscript.

A: We included a complete definition of the term "derecho" in the introduction.

3. P7L194-P7L200: Maybe outline if the environment was at least predicted correctly? Explain why the underestimation of wind speed cannot be attributed to the wind gust parametrization.

A: We added a sentence mentioning that the ECMWF-EPS underestimated the latent instability over Benelux (see Figure R3 below). The gust parametrizations in ECMWF and COSMO (with non-convection-permitting grid spacing) include a convective contribution to the simulated wind gust speed. Thus, if the environmental conditions do not favor deep convection, the model will not simulate a substantial contribution from the convection scheme.

[Figure]

**Figure R3.** Probability of CAPE being larger than 50 J/kg at 1800 UTC on 3 January 2014.

4. Figure 5: Why do you show fields from two different resolutions: (left) 7km and (right) 2.8 km (see above)?

A: We show the 7 km simulations for the pressure tendency and convergence to avoid a noisy visualization (we assume that the reviewer refers to Fig. 9). See also reply to major comment #3.

5. Figure 12: Add 2.8 km reflectivity?

A: We added the 2.8 km reflectivity to Figure 12 (see major comment #3).

**Technical Changes**

1. P3L70 – P3L86 These paragraphs need work. Your co-authors should be able to tell you how to make it more readable.

A: We removed some redundant information, and hope that the paragraph is more readable now.

2. P2L44 – P3L66: Reading a list of papers (a reader probably doesn't know) without much context does not motivate to continue reading. Put the literature in context, explain where the gap in research is and why you think it is interesting.

A: We outlined the European derecho literature a bit more.

3. P5L37 - P5L153: Although I enjoyed reading the detailed description of the evolution of cyclone Anne, there might be potential to shorten the text here.

A: To shorten this part, we removed some unnecessary information.

4. P5L154-P6L187 While it is certainly a good idea to spend a bit of time explaining convection precursors to NHESS readers, there might be some potential to shorten this part too.

A: Again, we tried to shorten this part by removing some unnecessary information.

5. P7L217 – P7220 From a technical perspective this is an interesting result you may want to highlight more. There is an ongoing discussion about resolution jump vs. LBC update frequency for limited area modeling. Also, specify the employed upper boundary condition (Relaxation or w=0?) in Section 2.

A: We thank the reviewer for this objection. Our intention was not to start a discussion on LBC frequency and/or resolution jump, because it is beyond the scope of this paper. This study is just an example that high-frequency LBCs are necessary for some events, since

otherwise the main forcing mechanism is missing. Additionally, we specified the employed upper boundary condition in Section 2 (relaxation is used).

Figures 2-7: In the beginning, 5 figures are shown to set the stage for the main ideas following. Are all of them needed? Along with the shortening potential in (3) and (4). There might be potential to remove some of the panels or move them to the supplement.

A: We agree with the reviewer that the number of figures could be reduced. Thus, we decided to remove Figure 5, since the statements that can be drawn are very general and do not demand an extra figure.

P1L11: trough favored→trough, and was favored

A: We replaced "trough favored" with "trough, and was favored".

P1L12: conditions were→environment was

A: We replaced "conditions" with "environment".

P1L14: You need to mention that these are limited-area simulations

A: We now mention limited-area simulations in the abstract.

P1L13: latent instability. Maybe use the more common term conditional instability (also rest of text)?

A: We use the term "latent instability" to address the availability of CAPE, since the term "conditional instability" only refers to the lapse rate profile, as pointed out by Schultz et al. 2000 (Schultz, D. M., P. N. Schumacher, and C. A. Doswell III, 2000: The intricacies of instabilities. Mon. Wea. Rev., 128, 4143-4148).

P1L15: datasets→reanalysis datasets

A: We added "reanalysis and operational" to "datasets".

P1L15: I would write "initial and lateral boundary conditions derived from ERA5"

A: We added "derived from ERA5".

P1L17: (i) convection-resolving scale→convection-resolving resolution (ii). At P1L14 you use the term convection-permitting. I would try and use just one of the two. We usually use convection-resolving, since the

A: We use now the term "convection-resolving" throughout the manuscript. Unfortunately, there is missing some text in the reviewers comment.

P1L18: This case study is testimony to the usefulness of ensembles of convection-resolving simulations to…

A: We included this expression into our sentence.

P1L21: affect←wrong word

A: We replaced "affect" with "occur".

P1L22: The style of the manuscript is unnecessarily cautious (hedging), which is legitimate to protect your claims, however, in most cases, it is actually not needed. For example: "In some cases, MCSs can exhibit". There is no need to add another can here. Check in the entire manuscript if vague language is really needed.

A: We removed "can" and other vague expressions throughout the manuscript.

P1L26: (i) remove: "which is" (ii) linearly organized→linearly-organized (also address hyphenation mistakes in rest of text)

A: We removed "which is" and addressed the hyphenation mistakes throughout the manuscript.

P1L28: remove: "region"

A: We removed the word "region".

P2L45: It is pointed out←rewrite

A: The expression "It is pointed out" is now replaced with "…, which showed that ...".

P2L50: AGL←define

A: AGL is now defined in the introduction.

P3L83: grid interval→grid spacing

A: We replaced "interval" with "spacing".

P4L95: realised→conducted

A: We replaced "realised" with "conducted".

P4L95: taken→derived

A: We replaced "taken" with "derived".

P4L110: Citing Baldauf et al (2011) may be warranted

A: We included Baldauf et al. (2011).

P5L138: intensive→deep

A: We replaced "intensive" with "deep".

P5L149: analysed→diagnosed

A: We replaced "analysed" with "diagnosed".

P7L206: "displaced" Wrong word, since in this Section, we don't know (yet) the true location.

A: We replaced "displaced" with "was located farther north".

P8L231: the highest-resolution run→the simulation with 1 km grid spacing

A: We replaced "the highest-resolution run" with "the simulation with 1 km grid spacing".

P8L235: "convection-initiating boundaries". Maybe choose a different term as it can be confused with the lateral boundaries, which you also discuss. Maybe "lifting mechanism"?

A: We replaced "boundaries" with "convergence zones".

P10L308-L310: Maybe mention that at least DWD and MCH employ such systems these days.

A: We have now included this information in the conclusions.

Table 1: Specifications about the physical parameterisations used in the different CCLM domains.→Simulation configurations

A: We changed the description of Table 1.

Figure 2 caption: I usually use the term "computational domain" (also in rest of text)

A: We replaced "model domain" with "computational domain" throughout the manuscript.

Figure 4: diagnosed 700 hPa upward motion.

A: We replaced "analysed" with "diagnosed".

---

## Referee Report (RR1)

**Review of the re-submitted version of "Synoptic-scale conditions and convection-resolving hindcast experiments of a cold-season derecho on 3 January 2014 in Western Europe" by Mathias et al.**

I did not find major issues not already mentioned in my initial review.

**Minor Comments**

Please re-consider all my technical comments from the initial review and make sure that they addressed consistently throughout the manuscript.

See for example:
P5L145: analysed → detected/diagnosed
P7L219 and P9275:  convection-initiating boundary (also pointed out by the anonymous reviewer)
P4L116 and P9L272 and P10L312: grid interval → grid spacing

**Technical Changes**
Fig 11: Panels (a) and (b) miss the "E" and "N" labels.

Please note that the manuscript is still missing a data availability section. In particular, I recommend including the following statement somewhere:  "COSMO may be used for operational and for research applications by the members of the COSMO consortium. Moreover, within a license agreement, the COSMO model may be used for operational and research applications by other national (hydro-)meteorological services, universities, and research institutes."

There is no need to mention my name in the acknowledgments.

Cheers
David Leutwyler

---

## Author Response (AR2)

**Referee #1**

**Review of the re-submitted version of "Synoptic-scale conditions and convection-resolving hindcast experiments of a cold-season derecho on 3 January 2014 in Western Europe" by Mathias et al.**

I did not find major issues not already mentioned in my initial review.

**Minor Comments**

Please re-consider all my technical comments from the initial review and make sure that they addressed consistently throughout the manuscript.

See for example:
P5L145: analysed → detected/diagnosed
A: "analysed" was replaced by "diagnosed".
P7L219 and P9275: convection-initiating boundary (also pointed out by the anonymous reviewer)
A: "convection-initiating boundary" was replaced by "convection-initiating zone" or "convection-initiating convergence zone" throughout the manuscript.
P4L116 and P9L272 and P10L312: grid interval → grid spacing
A: "interval" was replaced by "spacing" throughout the manuscript.

**Technical Changes**

Fig 11: Panels (a) and (b) miss the "E" and "N" labels.
A: The missing "E" and "N" labels were added.

Please note that the manuscript is still missing a data availability section. In particular, I recommend including the following statement somewhere: "COSMO may be used for operational and for research applications by the members of the COSMO consortium. Moreover, within a license agreement, the COSMO model may be used for operational and research applications by other national (hydro-)meteorological services, universities, and research institutes."
A: The statement about COSMO was added to the acknowledgements.

There is no need to mention my name in the acknowledgements.
A: The name was removed.

Cheers
David Leutwyler

**Referee #2**

The authors have addressed all points that I raised very well, however, major points 2 and 4 raised by reviewer 1 deserve at least some discussion in the manuscript and I recommend adding this discussion to the paper and / or appendix.

A: We added a sentence to section 2 regarding simultaneous horizontal and vertical resolution increase. Moreover, we reformulated the resolution sensitivity in section 4.2 and we clarified that the results of CCLM-ERA5 with 6-h updated lateral boundary conditions are similar to the ones obtained with CCLM-ECAN. We think that the changes made within the scope of the first major revision did already considerably improve the overall comprehensibility of sections 2 and 4.2.

[revised manuscript text omitted]